# PNSA, a Novel C-Terminal Inhibitor of HSP90, Reverses Epithelial–Mesenchymal Transition and Suppresses Metastasis of Breast Cancer Cells In Vitro

**DOI:** 10.3390/md19020117

**Published:** 2021-02-20

**Authors:** Aotong Zhang, Xin Qi, Fu Du, Guojian Zhang, Dehai Li, Jing Li

**Affiliations:** 1Key Laboratory of Marine Drugs, Chinese Ministry of Education, School of Medicine and Pharmacy, Ocean University of China, Qingdao 266003, China; zhangaotong@stu.ouc.edu.cn (A.Z.); qxhin@163.com (X.Q.); dufu@stu.ouc.edu.cn (F.D.); zhangguojian@ouc.edu.cn (G.Z.); dehaili@ouc.edu.cn (D.L.); 2Open Studio for Druggability Research of Marine Natural Products, Laboratory for Marine Drugs and Bioproducts, Qingdao National Laboratory for Marine Science and Technology, Qingdao 266237, China

**Keywords:** HSP90, inhibitor, EMT, metastasis, breast cancer

## Abstract

Metastasis accounts for the vast majority of deaths in breast cancer, and novel and effective treatments to inhibit cancer metastasis remain urgently developed. The expression level of heat shock protein 90 (HSP90) in invasive breast cancer tissue is higher than in adjacent non-cancerous tissue. In the present study, we investigated the inhibitory effect of penisuloxazin A (PNSA), a novel C- terminal inhibitor of HSP90, on metastasis of breast cancer cells and related mechanism in vitro. We found that PNSA obviously affected adhesion, migration, and invasion of triple-negative breast cancer (TNBC) MDA-MB-231 cells and Trastuzumab-resistant JIMT-1 cells. Furthermore, PNSA was capable of reversing epithelial–mesenchymal transformation (EMT) of MDA-MB-231 cells with change of cell morphology. PNSA increases E-cadherin expression followed by decreasing amounts of N-cadherin, vimentin, and matrix metalloproteinases9 (MMP9) and proteolytic activity of matrix metalloproteinases2 (MMP2) and MMP9. Comparatively, the N-terminal inhibitor of HSP90 17-allyl-17-demethoxygeldanamycin (17-AAG) had no effect on EMT of MDA-MB-231 cells. PNSA was uncovered to reduce the stability of epidermal growth factor receptor (EGFR) and fibroblast growth factor receptor (FGFR) proteins and thereby inhibiting their downstream signaling transductions by inhibition of HSP90. In addition, PNSA reduced the expression of programmed cell death-ligand 1 (PD-L1) to promote natural killer (NK) cells to kill breast cancer cells with a dose far less than that of cytotoxicity to NK cell itself, implying the potential of PNSA to enhance immune surveillance against metastasis in vivo. All these results indicate that PNSA is a promising anti-metastasis agent worthy of being studied in the future.

## 1. Introduction

Breast cancer is the most malignant and major cause of cancer-related deaths among women worldwide [1], which is classified into hormone-receptor-positive, human epidermal growth factor receptor-2 overexpressing (HER2+) and triple-negative breast cancer (TNBC) based on histological features. Therapy strategies vary according to the classification, including endocrine modulators, agents that target estrogen receptor (ER), and human epidermal growth factor receptor 2 (HER2) such as tamoxifen and trastuzumab. Much progress has been made in early detection and better treatment of breast cancer, leading to improved survival. However, a considerable number of patients will relapse as a result of organ metastasis, specifically lung, liver, bone, and brain [2]. It is reported that 20–30% of breast cancer patients may develop metastases after diagnosis and primary tumor treatment, and approximately 90% of cancer-related deaths are attributed to metastasis [3,4]. TNBC accounts for about 16% of all breast cancers, and is more aggressive than other breast cancer, and has a higher three-year recurrence rate and five-year mortality rate after treatment [5,6]. Due to the lack of estrogen receptor (ER), progesterone receptor (PR), and human epidermal growth factor receptor 2 (HER2), the conventional treatment used for other breast cancers is not effective in TNBC. In addition, the high heterogeneity of TNBC on the molecular level, pathology, and clinical characteristics also lead it to be hard to deal with [7]. HER2-positive tumors reach about 20–30% of breast cancers [8]. The application of HER2-directed antibodies, such as trastuzumab, has been suggested as standard therapy for HER2 positive advanced breast cancer. However, the resistance to anti-HER2 antibody has resulted in antibody drug-refractory metastatic and disease progression [9,10]. Therefore, novel and effective treatments to inhibit the metastasis of breast cancer cells remain urgently developed.

Epithelial–mesenchymal transition (EMT) is a process in which epithelial cells acquire mesenchymal features. In cancer, EMT is associated with tumor initiation, invasion, metastasis, and resistance to therapy. During EMT, epithelial cells undergo multiple morphologic, biochemical, and genetic rearranges that gradually enable them to acquire a mesenchymal phenotype [11,12]. The hallmark of EMT is the upregulation of N-cadherin followed by the downregulation of E-cadherin, and this process is regulated by a complex network of signaling pathways and transcription factors [13,14]. Reversal of EMT has been suggested as a valuable strategy for therapeutic intervention of metastatic cancer [15].

Heat shock protein 90 (HSP90) is a ubiquitous molecular chaperone protein and plays an important role in multiple biological functions and processes such as cell survival, proliferation, tumor progression, and metastasis by modulating the stability, maturation, and conformational changes of various proteins [16,17,18]. The expression level of HSP90 in invasive breast cancer tissue is higher than in adjacent non-cancerous tissue [19]. The elevated expression of HSP90 protein had a significantly positive connection with metastasis, advanced stage, and poor survival in patients [20]. In previous studies, we have reported that compound penisuloxazin A (PNSA), a new epipolythiodiketopiperazines (ETPs) from the mangrove endophytic fungus Penicillium janthinellum, is a novel C-terminal HSP90 inhibitor with the binding site at cysteine residues C572/C598 of HSP90 via disulfide bonds [21,22] (Figure 1), which is different from currently reported inhibitors that bind directly to HSP90 at ATP-binding pocket. The purity of PNSA was determined to be 98% (Appendix A) and its structure was verified by 1H NMR spectroscopy (Appendix A) [22]. Herein, we probe into the potential inhibitory effects of PNSA on EMTs and metastasis of triple-negative breast cancer cells and trastuzumab-refractory cells in vitro. Our results indicate that the novel inhibitor of HSP90 has the ability to reverse EMT and block cell migration and invasion.

## 2. Results

### 2.1. PNSA Impairs the Migration and Invasion of Breast Cancer Cells In Vitro

To evaluate the antimetastatic activity of PNSA, we first assessed the inhibitory effects of PNSA on the migration and invasion of MDA-MB-231 cells, triple-negative breast cancer cells, by the wound healing assay and the transwell assay. The results showed that PNSA obviously inhibited the two processes of cancer metastasis in a dose-dependent manner by PNSA. The Wound healing assay revealed that PNSA significantly decreased the migrations of MDA-MB-231 cells with the migration rates of 13.51% and 9.49% at 1, and 2 μM of PNSA, respectively (Figure 2A). Similarly, transwell assay results showed PNSA meaningfully inhibited invasive capabilities of MDA-MB-231 cells with inhibitory rates of 50.62% and 24.88% at indicated concentrations of PNSA (Figure 2C). 17-allyl-17-demethoxygeldanamycin (17-AAG) is a well-known N-terminal inhibitor of HSP90, we found that 17-AAG suppressed the migration and invasion of MDA-MB-231 cells dose-dependently. The migration rates were significantly decreased to 8.21% and 4.56%, and the invasion rates were 30.8% and 16.40% at 4, 8 μM of PNSA (Figure 2B,D). Within the range of dose in the experiments, PNSA and 17-AAG did not exhibit obviously cytotoxic activity as detected by sulforhodamine B (SRB) assay, which ruled out the influence of cell viability on cellular motility. (Figure 2E,F). Collectively, these results demonstrate that PNSA and 17-AAG all impair the migration and invasion of breast cancer cells.

### 2.2. PNSA Reverses EMT of MDA-MB-231 Cells

Previous studies have demonstrated that during the EMT process of cancer, intercellular interaction and the adherence of cancer cells to ECM components play important roles in the process of tumor metastasis [23]. As shown in Figure 3A, cellular polymers became larger with increasing concentrations of PNSA, the percentages of aggregated cells were significant compared to DMSO control at 28.80% and 45.93% with 1 and 2 μM of PNSA, indicating that PNSA promotes spontaneous cell aggregation (Figure 3A). In contrast, we found that PNSA inhibited attachment of MDA-MB-231 cells to coated Matrigels including many ECM components (Figure 3B). In addition, after PNSA treatment, the cell morphology gradually changed from spindle-shaped or polygonal mesenchymal morphology to flat polygonal epithelial-like cell morphology (Figure 3C). These results suggest that the PNSA may reverse the EMT process of MDA-MB-231 cells. N-cadherin is upregulated while E-cadherin is downregulated during an EMT in cancers, and this “cadherin switch” is associated with enhanced migratory and invasive traits [13]. Besides, EMT-related proteins vimentin and MMP9 were up-regulated, and the proteolytic activity of MMP was enhanced in tumor metastasis [24]. Meanwhile, C-Myc, a transcription factor, is also increased, strongly contributing to invasion and migration [25]. As shown in Figure 3D, PNSA induced a significant increase in E-cadherin expression and decreases in N-cadherin, vimentin, C-Myc, and MMP-9 expressions compared to DMSO group. Furthermore, the proteolytic activities of MMP-2 and MMP-9 in decomposing extracellular matrix were investigated after treatment of PNSA by gelatin zymography assay. As shown in Figure 3E, PNSA efficiently inhibited enzyme activities of MMP-2 and MMP-9 in MDA-MB-231 cells in a dose-dependent manner. For comparison, 17-AAG was revealed to have no influence on MDA-MB-231 morphology and E-cadherin protein level, although inhibiting N-cadherin expression (Figure 3C,D). These results indicate that PNSA with the new binding site on HSP90 has the ability to reverse EMT of MDA-MB-231 cells.

### 2.3. PNSA Inhibits Expressions and Activations of Signaling Molecules Related to EMT

It has been reported that multiple signaling pathways play a crucial role in the process of tumor metastasis such as PI3K/Akt, MAPK/ERK, JAK/STAT, and Wnt/β-catenin [26,27,28,29]. Our previous study showed that PNSA inhibited the metastasis of breast cancer at 12 h. In order to further explore the mechanism of PNSA underlying metastasis of MDA-MB-231 cells, we tested the effects of PNSA on several signaling pathways through Western blotting experiments at different concentrations and times. We found that PNSA significantly reduced active phosphorylated forms of ERK, Stat3, and Akt without changes in the total levels of these proteins. PNSA also downregulated the amounts of β-catenin and upstreaming membrane receptors in these signaling pathway epidermal growth factor receptor (EGFR), fibroblast growth factor receptor (FGFR)1 in MDA-MB-231 after 12 h treatment (Figure 4A). At 24 h, PNSA had more obvious effects on the expression levels or activations of these proteins (Figure 4B).

### 2.4. PNSA Inhibits the Protein Expressions and Activations in the Signaling Pathway by Targeting HSP90

In order to explore whether the declines in the protein expressions and activations as mentioned earlier were related to HSP90 inhibition, we further examined the influences of HSP90 on these proteins of MDA-MB-231 cells. MDA-MB-231 cells were treated with different fragments for siRNA-mediated depletion of HSP90 and related protein levels were determined by Western blotting. As shown in Figure 5A, Si-HSP90 was found to decrease protein levels of FGFR1, EGFR, C-Myc, and phosphorylation levels of ERK and AKT, but the amounts of N-Cadherin, ERK, AKT, and Stat3 were not changed in MDA-MB-231 cells. HSP90 inhibition induces degradation of its client proteins through the proteasome pathway [30]. We further investigated whether the down-regulation of these proteins were the result of proteasome degradation after PNSA treatment. EGFR and C-Myc were selected to be observed after MDA-MB-231 cells treated with the proteasome inhibitor MG132 before the addition of PNSA. We found that PNSA-induced reductions of EGFR and C-Myc were rescued by MG132 (Figure 5B). Furthermore, we found that PNSA treatment strongly protected HSP90 protein from destabilization at the indicated temperatures in MDA-MB-231 cells compared to DMSO control detected by cellular thermal shift assay (CETSA) method (Figure 5C). CETSA method directly detects the interaction between the drug molecule and target protein in cells, as the binding of the molecule can increase the thermal stability of protein [31]. Altogether, these results indicate that PNSA-induced down-regulations of these signaling molecules related to EMT are associated with protein degradations due to inhibition of HSP90. By the way, we noticed that HSP70 expression levels were not obviously affected by the treatment with PNSA. However, 17-AAG could induce an unfavorable increase of cytoprotective HSP70 expression (Figure 5D). Up-regulation of HSP70 was a striking disadvantage induced by HSP90 N-terminal inhibitors [32,33].

### 2.5. PNSA Inhibits the Migration, Invasion, and EMT-related Signaling Pathway Proteins in JIMT-1 Cells

Previous studies have shown that PNSA inhibits the metastasis of MDA-MB-231 cells which lack drug target receptors such as estrogen receptor (ER), progesterone receptor (PR), and human epidermal growth factor receptor 2 (Her2) [5]. To further explore the effect of compound PNSA on metastasis ability of other breast cancer cells, JIMT-1 cells, a trastuzumab-resistant form of breast cancer, were chosen to study further. As shown in Figure 6A,B, we found that PNSA suppressed the migration and invasion of JIMT-1 cells dose-dependently. PNSA at 0.5, 1, and 2 μM significantly decreased the migrations of JIMT-1 cells and the migration rates were 15.33%, 8.94%, and 6.60%, respectively (Figure 6A). Similarly, the transwell assay showed that PNSA meaningfully inhibited invasive capabilities of JIMT-1 cells with inhibitory rates at 56.75%, 40.08%, and 38.10% with indicated concentrations of PNSA (Figure 6B). No cell viability was affected by PNSA (Figure 6C). Next, we detected the influences of PNSA on the expressions and activations of several signaling proteins by Western blotting and the results showed that PNSA decreased active phosphorylated forms of ERK, Stat3, and Akt without changes in the total protein levels of these molecules, and downregulated the amounts of EGFR and FGFR1 at 12 h (Figure 6D). These results suggest that PNSA also inhibits metastasis of trastuzumab-resistant breast cancer cells in vitro.

### 2.6. PNSA Inhibits PD-L1 Protein Expression in MDA-MB-231 and Enhances NK Cells Cytotoxic Activity

Recent studies suggest that EMT processes induce upregulation of PD-L1 expression [34]. Interestingly, other reports also have confirmed that PD-L1 signaling plays an important role in maintaining the EMT state of various cancers including breast cancer [35,36]. Therefore, we further tested the effect of PNSA on PD-L1 protein level by Western blotting. As shown in Figure 7A, PNSA obviously downregulated the expression of PD-L1 protein in a dose-dependent manner. However, with immunoprecipitation analysis, PD-L1 was not found to combine with HSP90, which suggests that decrease of PD-L1 is not involving in HSP90 inhibition by PNSA and may be related to the reversal of EMT after PNSA treatment. (Figure 7B). 

Natural killer (NK) cells play a critical role in host immune responses against tumor growth and metastasis [37]. PD-L1/PD-1 is a pair of immune checkpoints, the reduction of PD-L1 protein expression can enhance the ability of NK cells to attack tumor cells by releasing more perforin and granzyme B^37^. We first digested the MDA-MB-231 cells, replanted them into the plates, and then added PNSA and NK92 cells for 5h to observe whether PNSA increased the killing ability of NK cells. Our results showed that NK92 cells exhibited enhanced-cytotoxicity activity against cancer cells after PNSA treatment compared with DMSO control at effector-to-target ratio T:E of 5:1 (Figure 7C). Furthermore, the releases of granzyme B and perforin were increased in PNSA-treated MDA-MB-231 cells (Figure 7D,E). During the time, no change of NK92 cell viability upon PNSA treatment was observed (Figure 7F). We further examined the effect of PNSA on NK92 cells viability for 72h and found that PNSA was much less cytotoxic to NK92 than to tumor cells with IC_50_ value at 12.96 μM, and the IC _50_ value of PNSA for MDA-MB-231 cells was 0.86 μM (Figure 7G). For comparison, the IC_50_ value of 17-AAG on NK92 cells for 72h was 9.08μM, meanwhile, the IC _50_ value for MDA-MB-231 cells was 3.46 μM (Figure 7H). These results suggest that PNSA may be better to trigger cancer immunology in vivo to deal with cancer metastasis.

## 3. Discussion

In the presence of the paper, we prove that PNSA, a novel c-terminal inhibitor of HSP90 with the binding site at cysteine residues C572/C598 of HSP90, has the potential to cope with metastasis of breast cancer cells, which is superior to 17-AAG, a well-known N-terminal inhibitor binding to ATP pocket of HSP90.

Breast cancer is a complex and heterogeneous disease. In recent years, various molecular targets are being explored including epidermal growth factor receptor (EGFR), poly(ADP-ribose) polymerase (PARP), and vascular endothelial growth factor (VEGF). Receptors, protein tyrosine kinases, phosphatases, proteases, PI3K/Akt signaling pathway, microRNAs (miRs), and long noncoding RNAs (lncRNAs) are potential therapeutic targets [38]. It has been reported that HSP90 empowers the evolution of resistance to target therapy in human breast cancer models and pharmacological inhibition of Hsp90 shows great promise in breast cancer treatment [39,40,41]. HSP90 consists of three different domains, an N-terminal ATP binding domain, middle domain, and C-terminal dimerization domain [17]. So far, most N- and C-terminal HSP90 inhibitors target the ATP-binding region of HSP90 to blocking ATP binding, thereby inhibiting the formation of HSP90–clients complex to promote degradations of these client proteins [42]. Inhibition of HSP90 can disrupt multiple signaling pathways that are important for the growth and metastasis of cancer cells [43]. However, the effects of different inhibitors of HSP90 on cancer metastasis and related mechanism are not very clear. Our data show that PNSA and 17-AAG all have the potential to inhibit migration and invasion of MDA-MB-231 cells. However, only PNSA is able to reverse EMT of MDA-MB-231 cells. We found that the cell morphology of MDA-MB-231 cells changed to epithelial cell shape obviously after treated with PNSA for 12 h. Furthermore, the molecule marker of epithelial cell E-cadherin was up-regulated while the molecule marker of mesenchymal cells N-cadherin, Vimentin, and MMP9 proteins PNSA were significantly down-regulated. However, 17-AAG did not obviously cause changes in E-cadherin protein expression and cell morphology. Our previous study found that slightly different binding [39] inhibitor-stirred HSP90 possesses selective inhibition potential on client proteins [22], we deduce that PNSA and 17-AAG may differently affect certain key proteins related to EMT development, which is worthy of being studied in the future. Increases in EMT process in primary breast cancer cells lead to enhance plasticity, enabling tumor progression, distant metastatic spread, and resistance to current chemotherapy and immunotherapy [44,45]. PNSA can reverse EMT of breast cancer cells and enhance the killing ability to breast cancer cells of NK cells by downregulation of PD-L1, which is better than 17-AAG, let alone PNSA is less toxic to NK cells than 17-AAG.

HSP90 is responsible for the maturation of more than 200 client proteins, and HSP90 inhibition usually induces degradation of its client proteins through the proteasome pathway [46]. The protein EGFR, FGFR, AKT, ERK, STAT3, vimentin, C-Myc, and MMP in our study are all reported to be client proteins of HSP90 (https://www.picard.ch/downloads/Hsp90interactors.pdf, accessed on 11 February 2021). Our results showed that in MDA-MB-231 cells, PNSA was able to bind to HSP90, and protein levels of EGFR, FGFR, vimentin, C-Myc, and MMP were reduced upon PNSA treatment. As an example, EGFR and C-Myc were also degraded via the proteasome pathway. However, the protein levels of AKT, ERK, and STAT3 were unchanged. With siRNA-mediated depletion of HSP90, we noticed that the amounts of AKT, ERK, and STAT3 were indeed not altered, which may due to the different cell lines to be used, as HSP90 inhibitor performs the inhibitory effect depending on cell context [22]. Other proteins such as N-cadherin and PD-L1 were not reported to be client proteins of HSP90. Consistently, protein levels of these two proteins were influenced by siRNA of HSP90, especially, PD-L1 was not associated with HSP90 by immunoprecipitation assay.

Multiple signaling pathways play a crucial role in the process of tumor metastasis [47,48]. EGFR frequent overexpression and/or hyperactivation in breast carcinomas play an active role in facilitating brain-specific metastatic spread [49,50]. In addition, FGFR aberrations increase the risk of brain metastases and predict poor prognosis in metastatic breast cancer patients [51,52]. We found that PNSA distinctly suppressed the protein levels of EGFR and FGFR, and subsequently inhibiting down-streaming signaling pathway with reduction of P-ERK, P-STAT3, and P-AKT. Wnt/β-catenin signaling contributes to EMT and metastasis in breast cancer. Deyet al. has found that the patients identified by the Wnt/β-catenin classifier had a greater risk of lung metastasis in TNBC [53]. Our data showed that the amount of β-catenin was decreased, and its target gene protein C-Myc was also reduced.

In summary, PNSA inhibits migration and invasion not only in TNBC cells but also in trastuzumab-resistant JIMT-1 cells. Furthermore, PNSA reverses EMT of breast cancer cells and promotes the cytotoxicity of NK cells. All these results indicate that the novel C-terminal inhibitor of HSP90 PNSA is a promising anti-metastasis agent worthy of being studied in the future.

## 4. Materials and Methods

### 4.1. Reagents

Dulbecco’s modified eagle’s medium (DMEM)-high glucose medium, Leibovitz’s L-15, and α-MEM were obtained from Gibco (Rockville, MD, USA). Fetal bovine serum (FBS) and Trypsin were obtained from Gibco-Invitrogen (Grand Island, NY, USA). Antibodies to detect N-cadherin MMP9, C-Myc, Vimentin, FGFR1, EGFR, Stat3, AKT, ERK c-Raf, β-catenin, heat shock protein 70 (HSP70), phosphorylated Akt (Ser473), phosphorylated Stat3 (Ser727), phosphorylated ERK were purchased from Cell Signaling Technology (Boston, MA, USA). PD-L1 and E-cadherin antibodies were purchased from Abcam (cambridge, UK). The primary antibodies (β-Tubulin, Actin, and GAPDH) and the secondary antibodies were purchased from HUABIO (Hangzhou, China). HSP90 antibody was purchased from Santa Cruz Biotechnology Inc. (Dallas, TX, USA). 17-allyl-17-demethoxygeldanamycin (17-AAG) was purchased from Apollo Scientific Limited (Stockport, UK). 3-(4, 5-dimethyl-2-thia-zolyl)-2, 5-diphenyl-2-H-tetrazolium bromide (MTT), sulforhodamine B (SRB) were purchased from Sigma-Aldrich (St. Louis, MO, USA). CCK-8, cell lysis buffer for Western blotting and IP, PMSF, MG132, and IgG antibody were purchased from Beyotime Institute of Biotechnology (Shanghai, China). Compound PNSA with purity 98% was obtained from Key Laboratory of Marine Drugs, Chinese Ministry of Education, School of Medicine and Pharmacy, Ocean University of China. The PNSA was analyzed by HPLC made by the HITACHI company equipped with a 5430 diode array detector and a C18 column (YMC-Pack ODS-A, 4.6 × 250 mm, 5 µm, 1 mL/min) by using stepwise gradient elution with 5–100% MeOH–H2O (0–5 min: 5%; 5–35 min: 5–100%; 35–45 min: 100%).

### 4.2. Cell Culture

MDA-MB-231, JIMT-1, NK92 cell lines were purchased from Shanghai Cell Bank. MDA-MB-231 cell line was cultured in Leibovitz’s L-15 with 15% bovine calf serum. JIMT-1 cell line was cultured in DMEM-High glucose medium with 10% bovine calf serum. NK92 cells were cultured in minimum essential medium alpha (α-MEM) containing heat-inactivated 12.5% horse serum (Solarbio, Beijing, China), 12.5% bovine calf serum, and 200 U/mL recombinant human IL-2 (Peprotech, Rocky Hill, CT, USA). All cell lines were grown to confluence at 37 °C in humidified atmosphere with 5% CO_2_.

### 4.3. Cell Proliferation Assay

MDA-MB-231 and JIMT-1 cells were evaluated by SRB assay. NK92 cells were measured by CCK-8 assay. Briefly, Logarithmic growing MDA-MB-231 and JIMT-1 cells were plated in 96-well plates at an initial density of 5 × 10^3^ per well. After 24 h, cells were treated with PNSA for 12 h or 72 h and SRB assay was used to evaluate the cell viability. The absorbance at 515 nm was measured using a microplate reader (BioTek, Winooski, VT, USA). NK92 cells were plated in 96-well plates at an initial density of 1 × 10^4^ per well. Cells were incubated with PNSA for 5 h or 72 h and CCK-8 assay was used to evaluate the cell viability. The absorbance value was detected by a microplate reader at 450 nm.

### 4.4. Migration and Invasion Assay

MDA-MB-231 and JIMT-1 cells were seeded in 96-well plates (5 × 10^4^ cells/well) and cultured for 24 h, obtaining a 90–100% confluent monolayer. Wounds were made with a p10 pipette tip and washed with PBS to eliminate non-adherent cells and cell debris, and a fresh medium with PNSA or DMSO was added. At 0 h and after 12 h, cells were photographed with a citation imaging reader. The empty area in each wound was quantified using Image J software (NIH) and compared to the corresponding initial wound. For invasion assay, the two kinds of breast cancer cells were tested with a Boyden chamber with 8 μm pore-size polycarbonate membranes coated with Matrigel (Corning). MDA-MB-231 and JIMT-1 (3 × 10^5^ cells/well) were seeded in a serum-free medium with the indicated concentration of PNSA in the upper chamber; the lower chamber contained 20% fetal bovine serum. After 12 h, the cells on the upper side of the membrane were removed with a cotton swab, and cells on the underside were fixed with methanol for 30 min and then stained with 0.1% crystal violet for 15 min. Photos were taken on an EVOS, and cells were counted with Image J Software. Migration (%) = [1 − (wound healing area at 12 h/wound healing area at 0 h)] × 100%. Invasion rate (%) = (the number of invading cells in PNSA group /the number of invading cells in DMSO group) × 100%

### 4.5. Cell Aggregation Assay

The cell aggregation assay was performed essentially as described previously [54]. Briefly, a total of MDA-MB-231(2.5 × 10^5^) cells in 1 mL of L-15 (serum-free) with different concentrations of PNSA were placed in polystyrene microtubes and shaken gently every 5 min for 1 h at 37 °C. Finally, glutaraldehyde (at a final concentration of 1% (*v/v*)) was used to stop the aggregation process. The percentage of aggregated cells was calculated as (1 − Ne/Nc) ×100%, where Ne is the number of single cells after incubation at 37 °C and Nc is the number of single cells before incubation.

### 4.6. Cell Adhesion Assay

MDA-MB-231 cells (1 × 10^4^ cells/well) were plated into 96-well plates. After 24 h, cells were treated with PNSA at different concentrations and incubated for 12 h. Then, cells were suspended in serum-free L-15 to form a single-cell suspension and were seeded into 96-well cell culture plates that had been precoated with Matrigel. After 45 min incubation at 37 °C, the wells were washed three times with PBS to remove non-adherent cells. 5mg/mL of MTT was added to each well for 4 h. After 4 h, the DMSO was added to each well and the absorbance value was detected by a microplate reader at 450 nm. Adhesion rate = (absorbance of PNSA group/absorbance of DMSO group) × 100%.

### 4.7. Gelatin Zymography Assay

MDA-MB-231 cells (3 × 10^6^ cells/well) were seeded in 6-well plates and the cells were allowed to attach for 24 h. The cells were then washed with PBS and incubated with serum-free medium containing PNSA for 12 h. Conditioned media from cell cultures treated with PNSA or DMSO were collected, centrifuged, and mixed with sample buffer (60 mM Tris-HCl pH 6.8, 25% glycerol, 2% SDS, 0.1% bromophenol) without reducing agents. The corresponding samples were loaded on 10% polyacrylamide gels with gelatin (1 mg/mL) and separated by gel electrophoresis. Gels were then washed with 2.5%Triton X-100 for 40 min and incubated in incubation buffer (50 mM Tris, 0.15 M NaCl, 10 mM CaCl_2_, 1 mM ZnCl_2_) at 37 °C for 36 h. The cell lysates derived from the same separation gel were transferred to a PVDF membrane (Bio-Rad, Hercules, CA, USA). The membranes were blocked in Rapid protein blocking solution for 20 min and then washed three times in TBST (Tris-buffered saline with 0.1% Tween-20). The membranes were incubated overnight at 4 °C with indicated primary antibodies and then probed with an HRP-conjugated secondary antibody for 1 h. Blots detected by chemiluminescence with the enhanced chemiluminescence (ECL) detection reagents.

### 4.8. Western Blotting

MDA-MB-231 cells were incubated with PNSA at indicated concentrations for 12 h, then the medium was removed, and cells were washed with ice-cold phosphate-buffered saline (PBS) twice, disrupted on ice for 35 min in loading buffer, and boiled for 15 min. Equal amounts of protein in cell lysates were separated by SDS-PAGE, followed by transferring to a PVDF membrane (Bio-Rad, Hercules, CA, USA). The membranes were blocked in Rapid protein blocking solution for 20 min and then washed three times in TBST (Tris-buffered saline with 0.1% Tween-20). The membranes were incubated overnight at 4 °C with indicated primary antibodies and then probed with an HRP-conjugated secondary antibody for 1 h. Blots were detected by chemiluminescence with the enhanced chemiluminescence (ECL) detection reagents.

### 4.9. Cell Transfection

RNA interference assay was performed as previously described [55]. MDA-MB-231 cells (3 × 10^5^ cells/well) were seeded in 6-well plates and the cells were allowed to attach for 24 h. The HSP90 si-RNA (Gene Pharma of Shanghai, Shanghai, China) were transfected into MDA-MB-231 cells for 48 h according to the manufacturer’s instructions. The sequences of siRNA were 5′-CCCUUCUAUUUGUCCCACGTT-3′ (#1187), 5′-GGACAGUUGGAAUUCAGAGTT-3′ (#1206), and 5′-CGUCUCGCAUGGAAGAAGU-3′ (#2042).

### 4.10. Cellular Thermal Shift Assay (CETSA)

MDA-MB-231 cells (3 × 10^5^ cells/well) were seeded in 6-well plates and the cells were allowed to attach for 24 h. After 24 h, cells were treated with PNSA at different concentrations and incubated for 3 h. After incubation, cells were resuspended in PBS and divided into several aliquots. Four aliquots were treated with DMSO and other four aliquots were treated with PNSA. Then, cells were heated at different temperatures by Biometra TOne PCR (Analytikjena, Jena, Germany). The heated cells were freeze-thawed three times with liquid nitrogen for 4 min every time and followed by centrifugation at 20,000× *g* for 20 min at 4 °C. The supernatants were harvested and loading buffer was added before boiling for 15min. Protein levels were assessed by Western blotting.

### 4.11. Immunoprecipitation

MDA-MB-231 cells were seeded in 6-well plates and the cells were allowed to attach for 24 h. After 24 h, the medium was removed, then washed twice with PBS, and disrupted on ice for 30 min. The lysates were centrifuged at 10,000× *g* for 15 min, and then the supernatants were harvested. Next, the lysates were incubated with 1 μg HSP90 antibody or control IgG overnight at 4 °C. The mixtures were incubated with a 40 μL slurry of protein A/G-agarose for 3 h at 4 °C. The immunoprecipitates were gathered by centrifugation and then washed six times with wash buffer (50 mM Tris-HCl, 150 mM NaCl, 1% Triton, pH 7.5). Finally, loading buffer was added and followed by boiling. Protein levels were assessed by Western blotting.

### 4.12. Cytotoxicity Assays of NK92 Cell

The cytotoxicity assay was performed essentially as described previously [56]. MDA-MB-231 cells (3 × 10^4^ cells/well) were seeded in 96-well plates and the cells were allowed to attach for 24 h. After 24 h, cells were treated with PNSA at different concentrations and incubated for 24 h. Then, we digested the MDA-MB-231 cells, replanted the plates, and after the cells adhered to the plates, added NK92 cells at an effector-to-target ratio of 5:1 for 5 h at 37 °C. After 5 h, 20 μL CCK-8 was added to each well, and the incubation continued for 1 h. The absorbance value was detected by a microplate reader at 450 nm. Cytotoxicity (%) = [1 − (experimental group OD/effector group OD)/ targeted group OD] × 100%

### 4.13. Analysis of Perforin and Granzyme by ELISA

MDA-MB-231 cells (3 × 10^4^ cells/well) were seeded in 96-well plates and the cells were allowed to attach for 24 h. After 24h, cells were treated with PNSA at different concentrations and incubated for 24 h. Then, we digested the MDA-MB-231 cells, replanted the plates, and after the cells adhered to the plates, added NK92 cells at an effector-to-target ratio of 5:1 for 8 h at 37 °C. After 8 h, supernatant was obtained from 96-well plates by centrifugation at 1500× *g* for 10 min at 4 °C and immediately frozen at −80 °C for further analysis. Perforin and Granzyme B levels in the serum were measured using human Perforin and Granzyme B ELISA kits (Dakewe Biotech Co, Ltd., Beijing, China), according to the manufacturer’s instructions.

### 4.14. Statistical Analysis

Data were presented as mean values ± standard deviation. Student’s *t*-test and one-way ANOVA analysis were used to analyze the differences between groups. Differences were considered statistically significant when *p* < 0.05.

## Figures and Tables

**Figure 1 marinedrugs-19-00117-f001:**
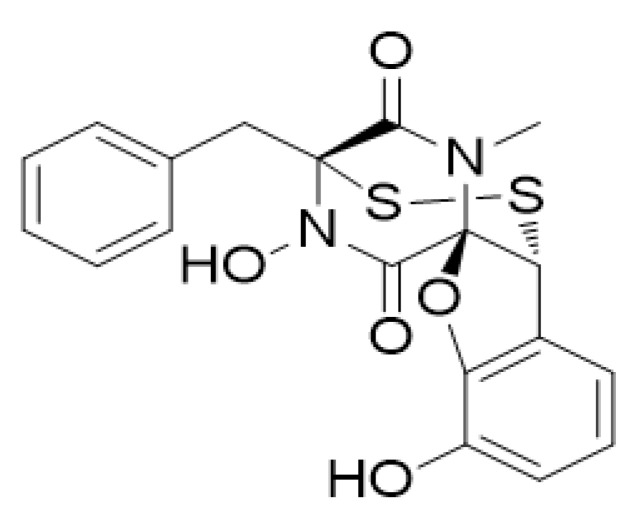
Chemical structure of penisuloxazin A (PNSA).

**Figure 2 marinedrugs-19-00117-f002:**
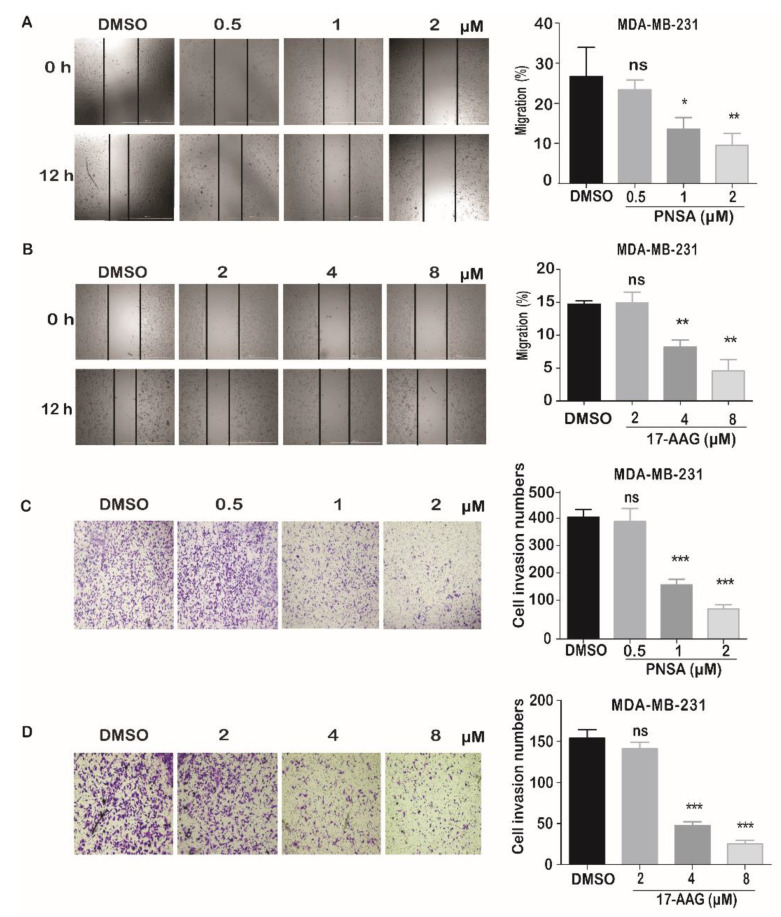
PNSA inhibits the migration and invasion of breast cancer cells. (**A**,**B**) Inhibition effects of PNSA and 17-allyl-17-demethoxygeldanamycin (17-AAG) on the migration of MDA-MB-231 cells. Representative images of the wound healing assay performed with MDA-MB-231 cells treated with indicated concentrations of PNSA (**A**) or 17-AAG (**B**), the migrated ratio was calculated (right panels). (**C**,**D**) Inhibition effects of PNSA and 17-AAG on the invasion of MDA-MB-231 cells. The MDA-MB-231 cells were incubated with the indicated concentrations of PNSA (**C**) or 17-AAG (**D**) for 12 h through a Matrigel-coated Boyden Chamber, the invasion cells were counted (right panels). (**E**,**F**) Effects of PNSA and 17-AAG on the viability of MDA-MB-231 cells. Cells were incubated with the indicated concentrations of PNSA (**E**) or 17-AAG (**F**) for 12 h and cell viability was measured using sulforhodamine B (SRB) assay. The bar graph represents the average ± SD of at least three independent experiments. * *p* < 0.05; ** *p* < 0.01; *** *p* < 0.001; ns, not significant (relative to DMSO-treated cells).

**Figure 3 marinedrugs-19-00117-f003:**
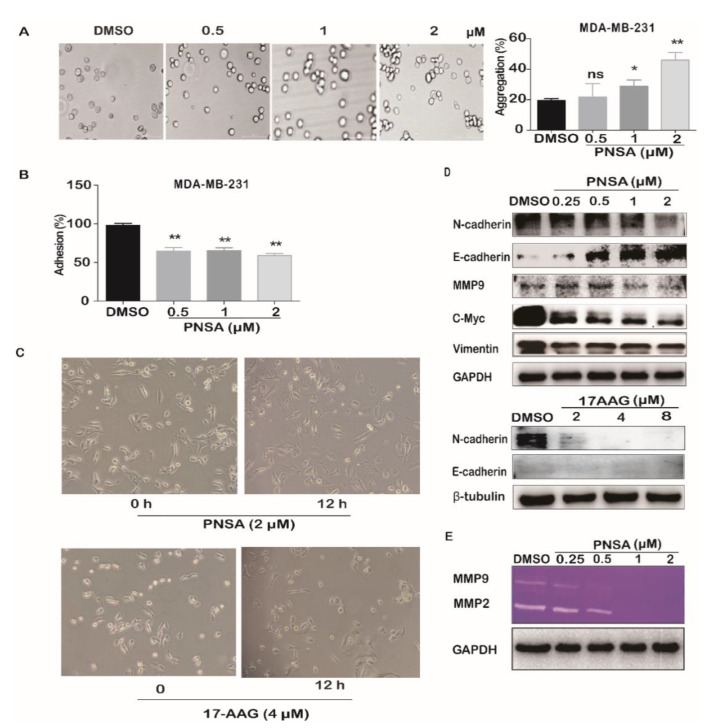
PNSA reverses epithelial–mesenchymal transformation (EMT) of MDA-MB-231 cells. (**A**) Effect of PNSA on cell aggregation. MDA-MB-231 cell suspensions were treated with different concentrations of PNSA (0.5, 1, 2 μM), and then cells were photographed (left panel) and counted for statistical analysis (right panel). (**B**) Effect of PNSA on cell adhesion. MDA-MB-231 cells were treated with different concentrations of PNSA (0.5, 1, 2 μM) for 12 h, the number of adhering cells was analyzed by 3-(4,5-dimethyl-2-thia-zolyl)-2,5-diphenyl-2-H-tetrazolium bromide (MTT) assay. (**C**) Effect of PNSA on cellular morphology. MDA-MB-231 cells were treated with PNSA (2 μM) (top panel) or 17-AAG (4 μM) (bottom panel) for 12 h and cell morphology was determined by a microscope. (**D**) Effects of PNSA and 17-AAG on the expressions of proteins related to metastasis. MDA-MB-231 cells were treated with PNSA (0.25, 0.5, 1, 2 μM) and 17-AAG (4 μM) for 12h. Protein levels were detected by Western blotting. (**E**) Inhibition effects of PNSA on proteolytic activities of MMP-9 and MMP-2. MDA-MB-231 cells were incubated with the indicated concentration of PNSA (0.25, 0.5, 1, 2 μM) for 12 h and proteolytic activities of MMP-9 and MMP-2 were measured by gelatin zymography assay. The bar graph represents the average ± SD of at least three independent experiments. * *p* < 0.05; ** *p* < 0.01; ns, not significant (relative to DMSO-treated cells).

**Figure 4 marinedrugs-19-00117-f004:**
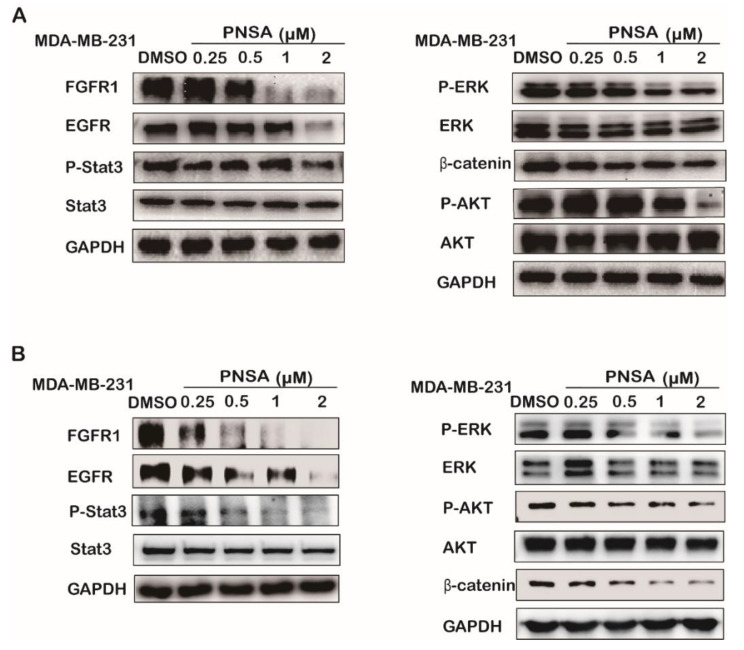
PNSA inhibits signaling molecules related to EMT analyzed by Western blotting. (**A**,**B**) Effects of PNSA on the expressions of signaling molecules related to EMT. MDA-MB-231 cells were treated with PNSA (0.25, 0.5, 1, 2 μM) for 12 h (**A**) or 24 h (**B**). GAPDH was used as a loading control.

**Figure 5 marinedrugs-19-00117-f005:**
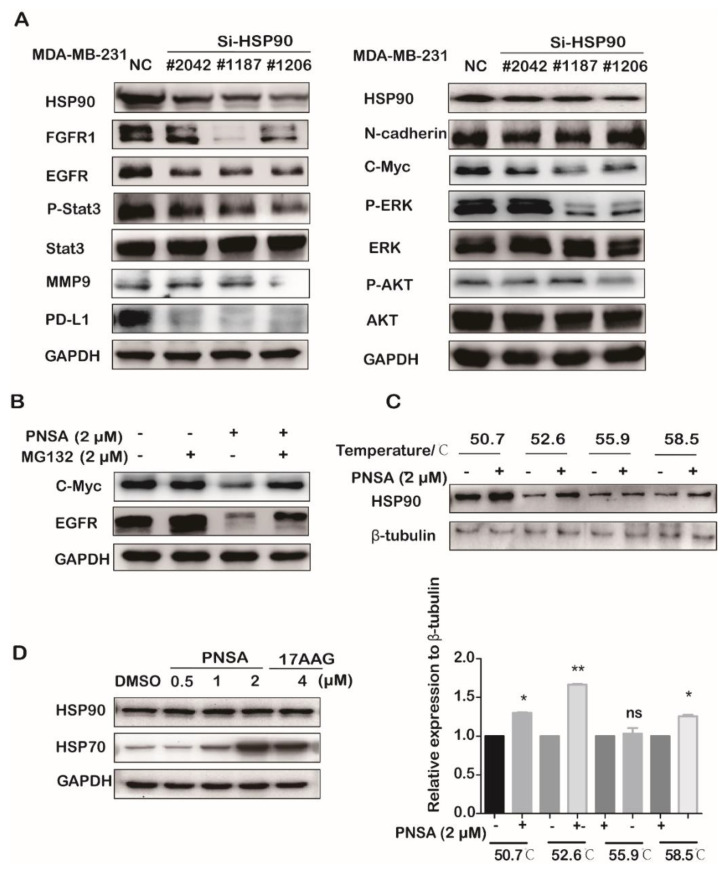
PNSA inhibits the protein expressions and activations in the signaling pathway by targeting heat shock protein 90 (HSP90). (**A**) Knockdown of HSP90 causes changes in protein expressions and activations. MDA-MB-231 cell line transfected with Si-RNA or Si-NC were cultured in 6-well plates for 48 h. Protein expressions and activations were assessed by Western blotting. (**B**) PNSA degrades HSP90 clients EGFR and C-Myc via proteasome pathway. MDA-MB-231 cells were treated with PNSA or/and MG132 (protease inhibitor) for 12 h. Protein levels were assessed by Western blotting. GAPDH was used as a loading control. (**C**) Effect of PNSA on HSP90 protein stabilization. MDA-MB-231 cells were treated with 2 μM PNSA for 3 h before heated at different temperatures. Western blotting was used to determine the protein levels (top panel). The protein band density was quantified by normalization to β-Tubulin (bottom panel). (**D**) Effects of PNSA and 17-AAG on the expressions of HSP70 and HSP90. MDA-MB-231 cells were treated with PNSA (0.5, 1, 2 μM) or 4 μM 17-AAG for 12 h. Proteins levels were analyzed by Western blotting. GAPDH was used as a loading control. The bar graph represents the average ± SD of at least three independent experiments. * *p* < 0.05; ** *p* < 0.01; ns, not significant (relative to DMSO-treated cells).

**Figure 6 marinedrugs-19-00117-f006:**
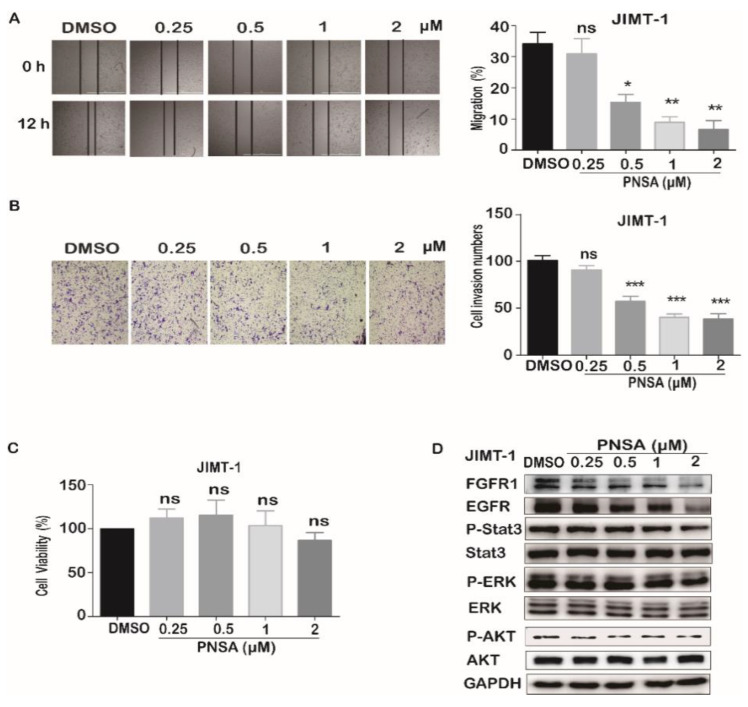
PNSA inhibits the migration, invasion, and EMT-related signaling pathway proteins of JIMT-1 cells. (**A**) Inhibitory effect of PNSA on the migration of JIMT-1 cells. Representative images of the wound healing assay performed with JIMT-1 cells treated with PNSA (0.25, 0.5, 1, 2 μM), the migrated ratio was calculated (right panel). (**B**) Inhibitory effect of PNSA on the invasion of JIMT-1 cells. JIMT-1 cells were incubated with PNSA (0.25, 0.5, 1, 2 μM) for 12 h through a Matrigel-coated Boyden Chamber, the invasion cells were counted (right panel). (**C**) Effect of PNSA on cell viability of JIMT-1. JIMT-1 cells were incubated with the indicated concentrations of PNSA for 12 h and cell viability was measured using SRB assay. (**D**) Effects of PNSA on the expressions and activations of signaling molecules related to EMT. Protein expressions and activations were analyzed by Western blotting. GAPDH was used as a loading control. The bar graph represents the average ± SD of at least three independent experiments. * *p* < 0.05; ** *p* < 0.01; *** *p* < 0.001; ns, not significant (relative to DMSO-treated cells).

**Figure 7 marinedrugs-19-00117-f007:**
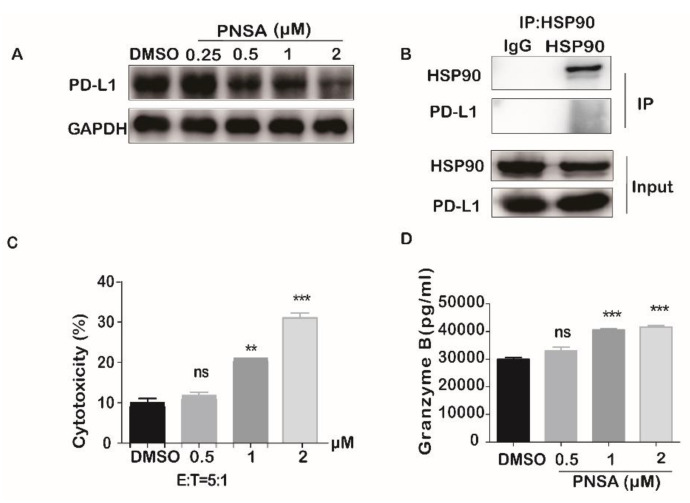
PNSA inhibits PD-L1 protein expression in MDA-MB-231 and enhances NK92 cells cytotoxic activity. (**A**) The effect of PNSA on the PD-L1 expression. MDA-MB-231 cells were treated with PNSA (0.25, 0.5, 1, 2 μM) for 24 h. Protein levels were analyzed by Western blotting. GAPDH was used as a loading control. (**B**) The association between HSP90 and PD-L1 protein. The HSP90 was immunoprecipitated from cell lysates using HSP90 antibody or IgG isotype (control) and immunoblotted with indicated antibodies. (**C**) PNSA treatment increases cytotoxic activity of NK92 cells on MDA-MB-231 cells. PNSA-treated MDA-MB-231 cells were co-cultured with NK92 cells at E: T ratio of 5:1, and cytotoxicity was measured as referred to in the Method. (**D**,**E**) PNSA treatment increases releases of granzyme B and perforin in NK92 cells; ELISA analysis of granzyme B and perforin levels in culture supernatant of NK92 cells when co-cultured with PNSA-treated MDA-MB-231 cells. (**F**) Effect of PNSA on cell viability of NK92. NK92 cells were incubated with the indicated concentrations of PNSA for 5h and cell viability was measured using CCK-8 assay. (**G**,**H**) Effects of PNSA and 17-AAG on cell viabilities of NK92 and MDA-MB-231 cells. Cells were incubated with the indicated concentrations of PNSA (**G**) or 17-AAG (**H**) for 72 h and cell viability was measured using SRB assay or CCK-8. The bar graph represents the average ± SD of at least three independent experiments. ** *p* < 0.01; *** *p* < 0.001; ns, not significant (relative to DMSO-treated cells).

## Data Availability

Not applicable.

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
