# Peer review of "PNSA, a Novel C-Terminal Inhibitor of HSP90, Reverses Epithelial–Mesenchymal Transition and Suppresses Metastasis of Breast Cancer Cells In Vitro"

_marinedrugs, 2021, doi:10.3390/md19020117_

Round 1

Reviewer 1 Report

Some in vitro pharmacological actions of the epipolythiodiketopiperazine alkaloid PNSA are reported in this study. In more detail, various invasion and metastasis-related biological endpoints have been investigated in MDA MB 231 and trastuzumab-resistant JIMT 1 breast cancer cell lines, and potentially interesting experimental results are reported. However, the manuscript is scattered with syntax and grammatical errors, hard to read in its actual form.

Author Response

Point 1. Some in vitro pharmacological actions of the epipolythiodiketopiperazine alkaloid PNSA are reported in this study. In more detail, various invasion and metastasis-related biological endpoints have been investigated in MDA MB 231 and trastuzumab-resistant JIMT 1 breast cancer cell lines, and potentially interesting experimental results are reported. However, the manuscript is scattered with syntax and grammatical errors, hard to read in its actual form.

Response 1: We feel great thanks for your professional review work on our article. As you are concerned, there are some grammatical errors that need to be addressed. We tried our best to improve the manuscript and made some changes to the manuscript. These changes will not influence the content and framework of the paper. And here we did not list the changes but marked in red in the revised paper. We appreciate for the reviewer warm work earnestly and hope that the correction will meet with approval.

Reviewer 2 Report

  • In this study Aotong Zhang 
et al., investigated the effect of PNSA, a novel C-terminal inhibitor of HSP90, on estrogen receptor negative breast (BCER-) . AA showed that PNSA inhibits migration and invasion in MDA-MB231 cancer cells. In addition AA showed that PNSA reverse the epithelial-mesenchymal transformation (EMT) of MDA-MB-231 cells

Major points were found that deserve attention by the authors:

  1. Abstract need to be re-written. There is No need to write in the abstract the methods (i.e. aggregation assay, zymography assay and so on ).
  2. The analysis of PNSA can be applied to other metastatic tumors? It is well worth to test this to make this finding more as a general mechanism shared by other metastatic tumors.
  3. Many evidence indicate that signal trasduction pathway such as PI3K play a key role in the pathogenesis of human disease such as a cancer. I suggest to study the activity of molecular signal transduction such as ERK or PI3K-Akt/PKB signal pathway (https://doi.org/10.1007/978-1-4939-1346-6_13), NOT only the downstream target as Akt/PKB phosphrylation. The Reviewer suggests a careful revision of the issue regarding the action on steroid receptor in breast cancer tumor. It is widely accepted the use of Hormonal (anti-estrogen) therapy works against hormone-receptor-positive breast cancer. Several findings reported that the molecular targets involved in the anti-growth activity may cause cycle arrest (https://doi.org/10.1016/j.steroids.2012.01.014) Tools that target steroid receptor in both hormone-dependent and hormone-independent phases should be preferred for their broader scope of use. In addition, specific signal transduction inhibitor may represent a useful tool against hormone-independent breast cancer. Many natural and chemical compounds may work targeting multiple signal transduction pathways and epigenetic modifications ( https://doi.org/10.1111/cpr.12022 https://doi.org/10.1016/j.semcancer.2019.04.006).

In conclusion this work has several elements of novelty with good correlations. However the AA should address the points indicated to make this work more convincing.

Author Response

Response to Reviewer 2 Comments

Point 1: Abstract need to be re-written. There is No need to write in the abstract the methods (i.e. aggregation assay, zymography assay and so on).

Response 1: Thank you for your nice comments on our article. According to your suggestions, we have rewritten the Abstract.

Point 2: The analysis of PNSA can be applied to other metastatic tumors? It is well worth to test this to make this finding more as a general mechanism shared by other metastatic tumors.

Response 2:  We thank you for your kind suggestion. Because metastasis accounts for the vast majority of deaths in breast cancer, and novel and effective treatments to inhibit the metastasis of breast cancer cells remain urgently developed. Therefore, in the present work, we pay more attention to breast cancer, especially triple negative breast cancer and trastuzumab-refractory breast cancer, which is more aggressive than other breast cancers due to lack of conventional treatment. In fact, we also studied the effect of PNSA on the metastasis of murine derived 4T1 cells in vitro, a kind of triple negative breast cancer cells, and found that PNSA impaired the migration and invasion of 4T1 cells. However,owing to milligram levels of compounds can be obtained from marine nature products, we have not performed anti-metastasis effect of PNSA in vivo yet. Yes, we will investigate whether PNSA can be applied to other metastatic tumors in future according to your recommendations.

Figure 1 is showed in Microsoft Word (Response to Reviewer 2 Comments)

Figure 1. PNSA inhibits the migration and invasion of 4T1 cells. (A) Inhibition effects of PNSA on the migration of 4T1 cells. Representative images of the wound healing assay performed with 4T1 cells treated with indicated concentrations of PNSA, the migrated ratio was calculated (right panel). (B) Inhibition effects of PNSA on the invasion of 4T1 cells. The 4T1 cells were incubated with the indicated concentrations of PNSA for 12 h through a Matrigel-coated Boyden Chamber, the invasion cells were counted (right panel). (C) Effects of PNSA on viability of 4T1 cells. Cells were incubated with the indicated concentrations of PNSA for 12 h and cell viability was measured using SRB assay. The bar graph represents the average ± SD of at least three independent experiments. *, P <0.05; **, P <0.01; ***, P <0.001; ns, not significant (relative to DMSO-treated cells)

Point 3: Many evidences indicate that signal transduction pathway such as PI3K play a key role in the pathogenesis of human disease such as a cancer. I suggest to study the activity of molecular signal transduction such as ERK or PI3K-Akt/PKB signal pathway (https://doi.org/10.1007/978-1-4939-1346-6_13), NOT only the downstream target as Akt/PKB phosphorylation. The Reviewer suggests a careful revision of the issue regarding the action on steroid receptor in breast cancer tumor. It is widely accepted the use of Hormonal (anti-estrogen) therapy works against hormone-receptor-positive breast cancer. Several findings reported that the molecular targets involved in the anti-growth activity may cause cycle arrest (https://doi.org/10.1016/j.steroids.2012.01.014) Tools that target steroid receptor in both hormone-dependent and hormone-independent phases should be preferred for their broader scope of use. In addition, specific signal transduction inhibitor may represent a useful tool against hormone-independent breast cancer. Many natural and chemical compounds may work targeting multiple signal transduction pathways and epigenetic modifications (https://doi.org/10.1111/cpr.12022 https://doi.org/10.1016/j.semcancer.2019.04.006).

Response 3: Yes, you are right, and thank you for your kind reminder. It has been reported that multiple signaling pathways play a crucial role in the process of tumor metastasis such as PI3K/Akt, MAPK/ERK, JAK/STAT and Wnt/β-catenin 1-3. In our manuscript, we have detected the changes of Akt (also called as PKB), STAT3 and their phosphorylation levels, and β-catenin in these signaling pathway. According to your suggestions, we have supplemented alterations of ERK and its phosphorylation level upon PNSA treatment in figure 3, figure 4A and figure 5D in the new version.

We have read the literatures you have provided carefully. Yes, there are many strategies to deal with breast cancers. As we all know, any kind of cancer is not a single disease, which is involving in different aberrant molecules in different signal transduction pathways related to proliferation and metastasis, including receptor or intracellular molecules. The aberrant molecule caused by abnormal expression due to epigenetic modifications or gene mutation can function as a target for development of anticancer drug. The molecules you have referred, such as steroid receptor, ERK, PI3K-Akt/PKB or epigenetic protein, are all targets for development of anticancer drugs. HSP90, an intracellular molecule, has been regarded as a promising target for development of anticancer drug. HSP90 is a ubiquitous molecular chaperone protein to modulate the stability, maturation and conformational changes of various proteins related to tumor progression and metastasis, including breast cancer4-10, therefore, Hsp90 acting as target to be explored its inhibitor is ever particularly appealing because it has the potential to simultaneously disrupt multiple pathways by acting on a single target11. Consistently, in our study, we found PNSA down-regulated various proteins in multiple signaling pathways, including PI3K/Akt, MAPK/ERK, JAK/STAT and Wnt/β-catenin. Hsp90 as a target for development of anticancer drug is valued by various pharmaceutical enterprises. As you requested, we have supplemented some descriptions in the Discussion.

Reference

  1. Brabletz, T.; Kalluri, R.;  Nieto, M. A.; Weinberg, R. A., EMT in cancer. Nat Rev Cancer 2018, 18 (2), 128-134.
  2. Takebe, N.; Warren, R. Q.; Ivy, S. P., Breast cancer growth and metastasis: interplay between cancer stem cells, embryonic signaling pathways and epithelial-to-mesenchymal transition. Breast Cancer Res 2011, 13 (3), 211.
  3. Zhu, X.; Bao, Y.;  Guo, Y.; Yang, W., Proline-Rich Protein Tyrosine Kinase 2 in Inflammation and Cancer. Cancers (Basel) 2018, 10 (5).
  4. Sankhala, K. K.; Mita, M. M.;  Mita, A. C.; Takimoto, C. H., Heat shock proteins: a potential anticancer target. Curr Drug Targets 2011, 12 (14), 2001-8.
  5. Schopf, F. H.; Biebl, M. M.; Buchner, J., The HSP90 chaperone machinery. Nat Rev Mol Cell Biol 2017, 18 (6), 345-360.
  6. Whitesell, L.; Lindquist, S. L., HSP90 and the chaperoning of cancer. Nat Rev Cancer 2005, 5 (10), 761-72.
  7. Zagouri, F.; Sergentanis, T. N.;  Nonni, A.;  Papadimitriou, C. A.;  Michalopoulos, N. V.;  Domeyer, P.;  Theodoropoulos, G.;  Lazaris, A.;  Patsouris, E.;  Zogafos, E.;  Pazaiti, A.; Zografos, G. C., Hsp90 in the continuum of breast ductal carcinogenesis: Evaluation in precursors, preinvasive and ductal carcinoma lesions. BMC Cancer 2010, 10, 353.
  8. Whitesell, L.; Santagata, S.;  Mendillo, M. L.;  Lin, N. U.;  Proia, D. A.; Lindquist, S., HSP90 empowers evolution of resistance to hormonal therapy in human breast cancer models. Proc Natl Acad Sci U S A 2014, 111 (51), 18297-302.
  9. Jhaveri, K.; Wang, R.;  Teplinsky, E.;  Chandarlapaty, S.;  Solit, D.;  Cadoo, K.;  Speyer, J.;  D'Andrea, G.;  Adams, S.;  Patil, S.;  Haque, S.;  O'Neill, T.;  Friedman, K.;  Esteva, F. J.;  Hudis, C.; Modi, S., A phase I trial of ganetespib in combination with paclitaxel and trastuzumab in patients with human epidermal growth factor receptor-2 (HER2)-positive metastatic breast cancer. Breast Cancer Res 2017, 19 (1), 89.
  10. Zagouri, F.; Sergentanis, T. N.;  Chrysikos, D.;  Papadimitriou, C. A.;  Dimopoulos, M. A.; Psaltopoulou, T., Hsp90 inhibitors in breast cancer: a systematic review. Breast 2013, 22 (5), 569-78.
  11. Chakraborty, A.; Koldobskiy, M. A.;  Sixt, K. M.;  Juluri, K. R.;  Mustafa, A. K.;  Snowman, A. M.;  van Rossum, D. B.;  Patterson, R. L.; Snyder, S. H., HSP90 regulates cell survival via inositol hexakisphosphate kinase-2. Proc Natl Acad Sci U S A 2008, 105 (4), 1134-9.

Reviewer 3 Report

The article presents research on a very interesting product that may become important in the treatment of TNBC in the future.

The description of the experiment was presented in a clear and understandable way

Author Response

Point 1: The article presents research on a very interesting product that may become important in the treatment of TNBC in the future. The description of the experiment was presented in a clear and understandable way.

Response 1: Thank you for your positive comments.

Round 2

Reviewer 2 Report

AA has improved the abstract.

On the line 164 "It has been reported that multiple signaling pathways play a crucial role in the process of tumor metastasis such as PI3K/Akt,MAPK/ERK,JAK/STAT and Wnt/β-catenin25-27." The refereces cited DO NOT refer to the above written sentence.

AA have to compare  specific PI3K inhibitor effects with PNSA inhibitor to highlight the molecolar mechanism.

Author Response

Response to Reviewer 2 Comments

Point 1: On the line 164 "It has been reported that multiple signaling pathways play a crucial role in the process of tumor metastasis such as PI3K/Akt, MAPK/ERK, JAK/STAT and Wnt/β-catenin25-27." The references cited DO NOT refer to the above written sentence.

Response 1: Thank you for carefully and patiently reviewing of our manuscript. We feel sorry for our carelessness. In our resubmitted manuscript, we have modified this mistake.

Point 2: AA have to compare specific PI3K inhibitor effects with PNSA inhibitor to highlight the molecular mechanism.

Response 2: We appreciate the reviewer’s insightful suggestion. However, in our study, we found that PNSA reduces the protein levels of FGFR and EGFR, and it plays an inhibitory role of Hsp90 in MDA-MB-231. As we know, Akt is the downstream signal molecule of PI3K, while FGFR and EGFR are the upstream of PI3K. Therefore, we think that PNSA inhibit the downstream signal molecule PI3K/p-Akt by inhibiting the molecular mechanisms of Hsp90. Therefore, it may not be necessary to compare with PI3K inhibitors. This report focuses on answering critical questions whether the new inhibitor of Hsp90 could inhibit the metastasis of breast cancer, which is different from the N-terminal inhibitor of Hsp90. We found our shortcomings in the present work. In the following study, we will compare with the drugs that have been reported, and it will contribute to the development and application of PNSA in cancer treatment strategies. We hope to get your understanding and we'd like to thank the reviewer for the careful readings and valuable comments.

Round 3

Reviewer 2 Report

AA should carry out the expts required to improve the MS rather than argue them.

Author Response

Response to Reviewer 2 Comments

Point 1: AA should carry out the experts required to improve the MS rather than argue them.

Response 1: We appreciate the reviewer’s insightful suggestion. Due to the outbreak, the university has forced students to have a holiday, and the time may not be appropriate. We hope to get your understanding and we feel sorry for the inconvenience brought to the reviewer.